# Empirical Bayes approach to Truth Discovery problems

**Tsviel Ben Shabat**[1]          **Reshef Meir**[1]          **David Azriel**[1]

[1]Dept. of Industrial Engineering and Management, Technion—Israel Institute of Technology

## Abstract

When aggregating information from conflicting sources, one's goal is to find the truth. Most real-value *truth discovery* (TD) algorithms try to achieve this goal by estimating the competence of each source and then aggregating the conflicting information by weighing each source's answer proportionally to her competence. However, each of those algorithms requires more than a single source for such estimation and usually does not consider different estimation methods other than a weighted mean. Therefore, in this work we formulate, prove, and empirically test the conditions for an Empirical Bayes Estimator (EBE) to dominate the weighted mean aggregation. Our main result demonstrates that EBE, under mild conditions, can be used as a second step of any TD algorithm in order to reduce the expected error.

## 1 INTRODUCTION

During the early 20-th century Sir Francis Galton, the 84 year old polymath, stumbled upon a prize winning contest where approximately 800 people paid a small fee to try and guess the weight of the presented live ox after it were to be slaughtered and dressed [Galton, 1907]. While no one had guessed the exact weight, Sir Francis noticed that the median guess had a negligible error. This seminal demonstration of "wisdom of the crowds" is an instructive example of *truth discovery*.

In a typical instance of *truth discovery* a group of workers answer questions that have correct yet unknown answers. One such question could be, "What is the height of the building in this image?" The workers who answer this question could be ordinary people, trained volunteers, a panel of experts, different computer algorithms, or a mix of all the above; all of whom we refer to as workers. One problem is

that some workers are better than others in some tasks, and some algorithms are better than others for different kinds of data. For example, if we know that worker *A*, is usually better at estimating buildings' heights than worker *B*, we would incorporate this fact into estimating the correct answer. For example, one can assign a different weight to different workers; in this example of the weighted mean method, it can be decided that the opinion of worker *A* weighs the same as the opinion of two workers. Like so, it can also be decided that the opinion of worker *B* weighs less than the opinion of a single worker, placing the weighted mean closer to the opinion of worker *A*.

Aitkin [1935] provided a formal solution to the problem of estimating the answer of a single numerical question, answered by multiple heterogeneous workers: if we know how competent each worker is, then is it optimal to weigh the worker proportionally to her level of competence (we will later explain these terms more formally). There are numerous works on estimating workers' competence which we discuss in Section 6.

For a single worker answering multiple questions, it may seem unlikely that there is any better solution than simply following the worker's answers. Yet, the Empirical Bayes approach shows that a better solution indeed exists [Stein, 1956, Casella, 1985].

In this paper we address the following question:

**Suppose we have *multiple questions* answered by *multiple heterogeneous workers*. Can the Empirical Bayes approach be exploited to improve upon existing truth-discovery algorithms?**

We explore this question both when workers' competence is known (in which case our baseline is Aitkin's estimator); and when the competence is *estimated*, using an arbitrary unbiased estimator of the true answers as baseline.

In Section 2 we formally cover the background we discussed above, that will be needed for the rest of the paper (in par-

*Accepted for the 38th Conference on Uncertainty in Artificial Intelligence* (UAI 2022).

ticular BLUE and Empirical Bayes).

We then turn to answer the question above using a simple principle: first aggregate workers' answers, and then apply an Empirical Bayes estimator on the outcome to improve it.

In the case of multiple workers with known competences who answer multiple questions (Section 3), we prove that our combined algorithm dominates the best linear unbiased estimator of Aitkin [1935].

In the more general setting where workers' competences are unknown (Section 4), we combine the Empirical Bayes estimator with an *arbitrary* (unbiased) truth discovery algorithm, and characterize an exact condition under which the combined algorithm improves upon the base algorithm. These results may also be of interest outside the truth-discovery domain, as an extension of the Empirical Bayes method to situations where the variance is estimated.

In Section 5, we demonstrate the benefit of the Empirical Bayes approach on synthetic and real datasets, by showing how it improves upon various truth-discovery algorithms from the literature that are used as a black-box, and discuss the practical conditions for such improvement.

Finally, we compare our work to some related papers in the truth discovery literature and discuss implications. Some proofs can be found in the full version
http://arxiv.org/abs/2206.04816.

## 2 PRELIMINARIES

Throughout the paper, we assume a set of $n$ workers provide answers to $m$ real-valued questions. The notation $\vec{1}$ denotes an $m$-length vector where all entries are '1'. For an $m$-length vector $\vec{v}$, we denote its mean by $\bar{v}$. An $n \times m$ matrix is denoted by a bold uppercase letter (e.g. $\boldsymbol{X}$).

### 2.1 MODEL AND NOTATION

**Noise Model** Unless mentioned otherwise, we assume workers answers follow the *additive white Gaussian* (AWG) noise model; see Diebold [1998]. Specifically, for a worker with variance $\sigma^2$, and a question whose true answer is $\mu$, the answer is a random variable sampled from the Normal distribution $\mathcal{N}(\mu, \sigma^2)$. That is, workers with lower $\sigma$ are more accurate.

**Observations** The response of the $i$-th worker to the $j$-th question is denoted by $X_{ij} \sim \mathcal{N}(\mu_j, \sigma_i^2)$; it is assumed that the responses are independent. Our goal is to estimate the $m$ unknown *ground truth* (GT) answers $\vec{\mu} = (\mu_1, .., \mu_m)$, We denote by $\vec{\sigma}^2 = (\sigma_1^2, \ldots, \sigma_n^2)$ the vector of workers' variances, where $\sigma_i^2$ is referred to as the inverse of the $i$-th workers' competences. It follows that competent workers have low variance and vice versa. As a concrete example,

one can think about the observations as a crowd-sourcing task where workers are presented with images of buildings and are told to estimate their heights. The number of workers is $n$ and the number of building images is $m$. The quantity $X_{ij}$ is the i-th worker estimate of building image $j$ and we wish to estimate the buildings' true heights $\mu$. The matrix of all responses is denoted by $\boldsymbol{X} \in \mathbb{R}^{n \times m}$ (with no subscript), the answers of $n$ workers to the j-th question is denoted by $\vec{X}_j = (X_{1j}, \ldots, X_{nj})$, and a dataset of a single worker as $\vec{X} \in \mathbb{R}^m$ and her variance is denoted by $\sigma^2$.

We denote by $\mathcal{P}_{\vec{\mu}, \vec{\sigma}}$ the distribution of $\boldsymbol{X}$ under the parameters $\mu, \sigma$, where $\vec{\mu} \in \mathbb{R}^m, \vec{\sigma} \in \mathbb{R}_+^n$, i.e, $\boldsymbol{X} \sim \mathcal{P}_{\vec{\mu}, \vec{\sigma}}$ (and $\mathcal{P}_{\vec{\mu}, \vec{\sigma}}$ follows the AWG model unless stated otherwise) . We denote by $E_{\vec{\mu}, \vec{\sigma}}[\cdot]$ and $Var_{\vec{\mu}, \vec{\sigma}}[\cdot]$ the expected value and variance of the term in brackets, respectively, for given parameters. That is, $E_{\vec{\mu}, \vec{\sigma}}[\cdot]$ is a shorthand for $E_{\boldsymbol{X} \sim \mathcal{P}_{\vec{\mu}, \vec{\sigma}}}[\cdot]$ and likewise for the variance.

**Algorithms** A *truth discovery algorithm* is a function $A : \mathbb{R}^{n \times m} \to \mathbb{R}^m$, mapping an observation matrix to a vector of estimated answers.[1] An algorithm may also take additional information as input. In particular, a *variance-based algorithm* (denoted by $A^\sigma$) is assumed to have access to the true variance of each worker.

**Evaluation** Given a *truth discovery algorithm*, we are interested in how far $A(\boldsymbol{X})$ is from the true answers $\vec{\mu} \in \mathbb{R}^m$, in expectation.

Formally, we denote by $\mathcal{L}(\hat{\mu}, \vec{\mu})$ the *loss* of estimation $\hat{\mu} \in \mathbb{R}^m$. Throughout this work, the loss function $\mathcal{L}$ is the square euclidean norm, i.e, $\mathcal{L}(\hat{\mu}, \vec{\mu}) := \|\hat{\mu} - \vec{\mu}\|_2^2$. We then measure the loss of A on a particular input as $\mathcal{L}(A(\boldsymbol{X}), \vec{\mu})$.

Because the observations $\boldsymbol{X}$ are random, we will use the expected loss. Formally, the *risk* of Algorithm A (given parameters $\vec{\mu}, \vec{\sigma}$) is

$$\mathcal{R}_{\vec{\mu}, \vec{\sigma}}(A) := E_{\vec{\mu}, \vec{\sigma}}[\mathcal{L}(A(\boldsymbol{X}), \vec{\mu})].$$

Our goal then is to find some algorithm A with low risk, i.e, to minimize $\mathcal{R}_{\vec{\mu}, \vec{\sigma}}(A)$ for every $\vec{\mu}$ and $\vec{\sigma}$.

### 2.2 THE BEST LINEAR UNBIASED ESTIMATOR (BLUE)

Recall that an estimator of a parameter is:

- *unbiased* if its expected value equals the estimated term;
- *linear* if it is a linear function of the observations.

---

[1] This is sometimes called an estimator but since we consider various types of estimators in this work, we use the term truth discovery algorithm to avoid confusion.

| $X_{ij}$ | 1 | 2 | 3 | 4 | $\sigma_i^2$ |
|---|---|---|---|---|---|
| 1 | 20 | 2 | 3 | 4 | 93.5 |
| 2 | 10 | 11 | 18 | 14 | 11 |
| 3 | 8 | 11 | 23 | 19 | 34.5 |
| 4 | 6 | 13 | 7 | 3 | 56.5 |
| $GT$ | 10 | 9 | 12 | 16 | $\mathcal{L}$ |
| $AVG$ | 11 | 9.25 | 12.75 | 10 | 9.41 |
| $A_B^\sigma$ | 9.85 | 10.6 | 16.6 | 12.95 | 8.22 |
| $\text{EBBLUE}^\sigma$ | 10.25 | 10.82 | 15.5 | 12.65 | 6.68 |

**Table 1:** An example of a data set, $X_{ij}$ is the i-th worker response for the j-th question. $\sigma_i^2$ is the calculated variance of the i-th worker relative to the ground truth (GT), $\mathcal{L}$ is the loss of each estimator

We denote by $\Delta_{LUE}$ the set of all linear unbiased estimators $A^\sigma$, i.e. all linear unbiased truth discovery algorithms that also have access to workers' variance.

Consider the following estimator/algorithm:

$$A_B^\sigma(\vec{X}_j) := \Big(\sum_{i=1}^n \frac{1}{\sigma_i^2}\Big)^{-1} \sum_{i=1}^n \frac{X_{ij}}{\sigma_i^2}.$$

and $A_B^\sigma(\boldsymbol{X}) := (A_B^\sigma(\vec{X}_j))_{j \leq m}$. It can be easily shown that $A_B^\sigma$ is an unbiased estimator i.e $E_{\vec{\mu},\vec{\sigma}}[A_B^\sigma(\boldsymbol{X})] = \vec{\mu}$.

**Theorem 2.1** ([Aitkin, 1935]). *Under the AWG model, $\mathcal{R}_{\vec{\mu},\vec{\sigma}}(A^\sigma) \geq \mathcal{R}_{\vec{\mu},\vec{\sigma}}(A_B^\sigma)$, for all $A^\sigma \in \Delta_{LUE}, \vec{\mu} \in \mathbb{R}^m, \vec{\sigma} \in \mathbb{R}_+^n$.*

In words, the theorem of Aitkin [1935] shows that the inverse variance weighing of the observations is the best linear unbiased estimator (BLUE) for $\vec{\mu}$ under the square loss function. In particular, the BLUE uses the input on each question separately, and thus the risk is independent of the number of questions $m$. As per our buildings' heights example, if we know how competent each worker is (at estimating buildings' heights from images), the best unbiased linear way of estimating the real height is a weighted average of the workers' answers, where the weight of worker $i$ is $1/\sigma_i^2$.

Table 1 is an example for a dataset, where workers' variances are known and the loss for this particular instance is compared between different estimators.

## 2.3 THE EMPIRICAL BAYES ESTIMATOR (EBE)

In a seminal paper, Stein [1956] introduced an estimator that dominates BLUE in a simple estimation problem of a normal distribution. In the current setting, in the case of a single worker estimating multiple questions, Stein's result implies that estimating the true answers using the input of the worker directly, is dominated by another estimator, which is rather surprising. Consequently, different variations of Stein's estimator and its derivation from empirical Bayesian statistics perspective are introduced by Efron and Morris [1973]. Here we focus on one such version.
To define the estimator, we consider a setting of a single worker with responses $\vec{X} = (X_1, \ldots, X_m)$, following a

single-worker AWG model. That is, $X_1, \ldots, X_m$ are independent with $X_j \sim \mathcal{N}(\mu_j, \sigma^2)$, $j = 1, \ldots, m$.

A *modifying estimator* is a function $\phi : \mathbb{R}^m \times \mathbb{R}_+ \to \mathbb{R}^m$, which can 'modify' a vector of responses. The estimator can also accept an additional parameter, which we can think of as the (true or estimated) variance. As with truth discovery algorithms, we denote by $\phi^\sigma : \mathbb{R}^m \to \mathbb{R}^m$ modifying estimators that have access to the true variance $\sigma^2$.

A trivial example is the identity estimator $\phi_I(\vec{X}) := \vec{X}$.

**Definition 2.1** (Empirical Bayes Estimator).

$$\phi_{\text{EB}}(\vec{X}, \sigma) := \bar{X}\vec{1} + \Big[1 - \frac{(m-3)\sigma^2}{\|\vec{X} - \bar{X}\vec{1}\|^2}\Big](\vec{X} - \bar{X}\vec{1}). \quad (1)$$

Arranged differently, $\phi_{\text{EB}}(\vec{X}, \bar{\sigma}) = \bar{X}\vec{1}\frac{(m-3)\sigma^2}{\|\vec{X}-\bar{X}\vec{1}\|^2} + \vec{X}\Big[\vec{1} - \frac{(m-3)\sigma^2}{\|\vec{X}-\bar{X}\vec{1}\|^2}\Big]$ is a weighted average of each component of $\vec{X}$ and its mean $\bar{X}$. It is also instructive to notice that when $m = 3$ then $\phi_{\text{EB}}(\vec{X}, \sigma) = \vec{X}$, i.e, $\phi_{\text{EB}}$ becomes the identity estimator $\phi_I$, and when $m$ goes to infinity we get that $\lim_{m \to \infty} \frac{(m-3)\sigma^2}{\|\vec{X} - \bar{X}\vec{1}\|^2} = \frac{\sigma^2}{\sigma^2 + C}$ where $C \in \mathbb{R}_+$ is a constant related to the variance of the ground truth. We provide an explicit expression for C, and the derivation of the EBE in the full version.

**Theorem 2.2** ([Lehmann and Casella, 1998]). *In the AWG model with a single worker and $m > 3$ questions,*

$$\mathcal{R}_{\vec{\mu},\sigma}(\phi_{\text{EB}}^\sigma) < \mathcal{R}_{\vec{\mu},\sigma}(\phi_I) \text{ for all } \vec{\mu} \in \mathbb{R}^m, \sigma \in \mathbb{R}^+. \quad (2)$$

In words, the empirical Bayes estimator for $\vec{\mu}$, which is not linear and not unbiased (but has access to the true variance $\sigma$), has strictly lower risk than $\phi_I$. Note that for a single worker, $\phi_I$ coincides with the BLUE $A_B^\sigma$.

We also consider Stein's estimator, which is defined now.

**Definition 2.2** ([Stein, 1956]).

$$\phi_{Stein}(\vec{X}, \sigma) := \Big[1 - \frac{(m-2)\sigma^2}{\|\vec{X}\|^2}\Big]\vec{X}$$

Stein estimator can be thought of as an empirical Bayes estimator with a normal prior and where the prior's mean is known to be 0.

## 3 KNOWN COMPETENCE

We now return to the general multi-worker, multi-question setting.

We begin by assuming that we know the competences of the workers $\sigma_i$ for $i = 1, \ldots, n$, an assumption that will be relaxed later.

We will use the following result which stems from Neyman-Fisher factorization theorem. See full version for the proof.

**Algorithm 1:** $\text{EBBLUE}^\sigma$ for Known Competence

**Input:** Dataset $\boldsymbol{X} \in \mathbb{R}^{n \times m}$, variances $\vec{\sigma} \in \mathbb{R}_+^n$

$\vec{X}^B \leftarrow A_B^\sigma(\boldsymbol{X})$ ;

$\tilde{\sigma}^2 \leftarrow \left( \sum_{i=1}^n \frac{1}{\sigma_i^2} \right)^{-1}$;

**return** $\phi_{\text{EB}}(\vec{X}^B, \tilde{\sigma})$ ;

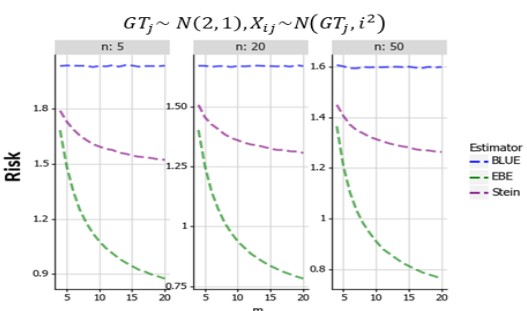

**Figure 1:** Each data point is a 100,000 samples average, each sample includes new GT and new workers.

**Proposition 3.1.** $A_B^\sigma(\boldsymbol{X})$ *is a sufficient statistic for* $\vec{\mu} = (\mu_1, .., \mu_m)$ *under* $\mathcal{P}_{\vec{\mu}, \vec{\sigma}}$.

It follows that there is no loss of relevant information when considering only $A_B^\sigma(\boldsymbol{X})$ instead of the observation matrix. That is, we can replace our observations $\boldsymbol{X} \in \mathbb{R}^{n \times m}$ with a single 'aggregated worker" who answered $m$ questions, denoted by $\vec{X}^B = (X_1^B, ..., X_m^B) = A_B^\sigma(\boldsymbol{X})$. The variance of this single worker (denoted by $\tilde{\sigma}^2$) is the harmonic mean of all workers' variances, divided by $n$.

Our first algorithm $\text{EBBLUE}^\sigma$ (see Alg. 1) simply uses Aitkin's BLUE to aggregate the labels independently on each question, then applies the Empirical Bayes estimator on the outcome. Since $\tilde{\sigma}^2$ is the true variance of the aggregated worker, $\phi_{\text{EB}}(\vec{X}^B, \tilde{\sigma}) = \phi_{\text{EB}}^\sigma(\vec{X}^B)$, so intuitively we are back to applying Empirical Bayes in a single-worker scenario. Indeed, from Proposition 3.1 and Theorem 2.2 we get our main result for this section:

**Corollary 3.1.1.** *In the AWG model, for* $m > 3$ *and any* $n$,

$$\mathcal{R}_{\vec{\mu}, \vec{\sigma}}(\text{EBBLUE}^\sigma) < \mathcal{R}_{\vec{\mu}, \vec{\sigma}}(A_B^\sigma) \text{ for all } \vec{\mu} \in \mathbb{R}^m, \vec{\sigma} \in \mathbb{R}_+^n \tag{3}$$

In words, Corollary 3.1.1 says that if we know the workers' competences then Alg. 1 strictly beats the unmodifed BLUE, and by Theorem 2.1 strictly beats any unbiased linear estimator. In statistical terms, BLUE (and any other unbiased linear estimator) is an inadmissible estimator.

Figure 1 demonstrates inequality (3). We generated ground truth $\mu_j \sim N(2,1)$, $j = 1, \ldots, m$ independently, and noisy observations $X_{ij} \sim N(\mu_j, i^2)$, then we aggregated the answers into a single worker using $A_B^\sigma$, applied different modifying estimators on the outcome, and calculated their (empirical) risk over 4 samples. It can be seen that the advantage

of EBE and Stein is more significant when fewer workers answer more questions.

# 4 ESTIMATED COMPETENCE

Often in real world problems we have no access to the true variances of our workers, $\vec{\sigma} \in \mathbb{R}_+^n$. However we can estimate it from the observations. Many *truth discovery* algorithms are doing exactly that—either in a supervised way (if we have access to the true answers of some questions) or unsupervised (by comparing workers to one another).

We therefore abstract away from our Alg. 1, by assuming we only have access to an estimate of the aggregated worker's variance. Crucially, our analysis is oblivious as to *how* the variance is estimated, or to how observations were aggregated, and may therefore apply for any truth-discovery algorithm.

Similarly to the previous section, we consider the aggregated answer vector. However rather than $A_B^\sigma$ (which requires the actual workers' variances), we now assume an arbitrary truth discovery algorithm $A : \mathbb{R}^{n \times m} \to \mathbb{R}^m$ is used, together with some estimator of the variance $\psi : \mathbb{R}^{n \times m} \to \mathbb{R}_+$.

Then, our general $\text{EB}_A^\psi$ algorithm (see Alg. 2) simply applies Eq. (1) to modify the output of algorithm A, using the estimated variance $\hat{\sigma}^2 = \psi(\boldsymbol{X})$.

**Algorithm 2:** $\text{EB}_A^\psi$ for estimated Competence

**Input:** Dataset $\boldsymbol{X} \in \mathbb{R}^{n \times m}$

$\vec{X}^A \leftarrow A(\boldsymbol{X})$;

$\hat{\sigma}^2 \leftarrow \psi(\boldsymbol{X})$;

**return** $\phi_{\text{EB}}(\vec{X}^A, \hat{\sigma})$;

Our analysis is divided into two parts: we first analyze the risk of Alg. 2 under the minimal assumption that each $X_j^A$ is an unbiased estimator of $\mu_j$ in Section 4.1. Then, Section 4.2 considers the case where the answers of the aggregated worker are assumed to be normally distributed around the true answers.

## 4.1 GENERAL MODEL

As previously mentioned, *truth discovery* algorithms typically estimate the truth by estimating workers' competence and then aggregate answers, weighing them accordingly.

In general we may not know the distribution of the aggregated answer, either since the initial observations depart from the AWG model, or because the algorithm A is complicated or unknown. We thus relax any assumption on the input in this section, except that $\vec{X}^A$ is *unbiased*. Thus $\vec{\mu} = E[\vec{X}^A]$. We also denote the true (unknown) variance by $\sigma^2 := Var[\vec{X}_j^A]$ (identical for all $j$). We next present a

sufficient condition for our Alg. 2 to have smaller risk than its baseline algorithm A for all $\mu \in \mathbb{R}^m$.

**Theorem 4.1.** *For any unbiased algorithm* A, *and* $m > 3$,[2]

$$\mathcal{R}_{\vec{\mu}}(\text{EB}_A^\psi) < \mathcal{R}_{\vec{\mu}}(A) \text{ for all } \vec{\mu} \in \mathbb{R}^m$$

*if and only if*

$$2(m-3)\Sigma_{j=1}^m Cov\Big(X_j^A, \frac{\psi(\boldsymbol{X})(X_j^A - \bar{X}^A)}{\|\vec{X}^A - \bar{X}^A\vec{1}\|^2}\Big)$$

$$- (m-3)^2 E_{\vec{\mu}}\Big(\frac{(\psi(\boldsymbol{X}))^2}{\|\vec{X}^A - \bar{X}^A\vec{1}\|^2}\Big) > 0. \quad (4)$$

The condition in Theorem 4.1 may seem somewhat obtuse, yet we argue it may still be useful:

- It is easy to see that by choosing $\psi(\vec{X})$ that is sufficiently close to 0, the condition holds;

- The condition is purely a function of the observations $\boldsymbol{X}$, therefore it can be verified empirically, given enough samples;

- It leads to an improvement of the algorithm, as we explain below;

- Under additional assumptions on the distribution, the condition is substantially simplified and provides important intuition (Section 4.2).

The main assumption of Theorem 4.1 is that the output of the truth discovery algorithm used as baseline is unbiased, i.e., $E(\vec{X}^A) = \vec{\mu}$. This assumption may not hold, when workers provide biased estimates. For example, in the buildings heights setting, if workers systematically overestimate the buildings' heights, then the assumption is violated (see proof in full version).

**Generalizing further?** While the modifying estimator $\phi_{\text{EB}}$ we apply is optimal for a *single worker*, it turns out that we can do better in the multi-worker case.

We define the *generalized EB estimator* $\phi_{\text{EB}}^\alpha$ by replacing the $(m-3)$ term in Def. 2.1 with $\alpha \in \mathbb{R}_+$; and denote by $\text{EB}_A^{\psi,\alpha}$ the corresponding generalized version of Alg. 2.

**Proposition 4.2.** $\mathcal{R}_{\vec{\mu}}(\text{EB}_A^{\psi,\alpha})$ *is minimized by setting*

$$\alpha^* := \frac{\Sigma_{j=1}^m Cov\Big(X_j^A, \frac{\psi(\boldsymbol{X})(X_j^A - \bar{X}^A)}{\|\vec{X}^A - \bar{X}_A\vec{1}\|^2}\Big)}{E\Big(\frac{\psi(\boldsymbol{X})^2}{\|\vec{X}^A - \bar{X}^A\vec{1}\|^2}\Big)}.$$

Proof is in the full version.

---

[2]Since in this subsection we do not assume that the distribution of $\boldsymbol{X}$ follows the AWG model, we do not need a parameter for the individual competence. Other than that, all definitions remain the same.

## 4.2 NORMAL MODEL

Testing whether inequality (4) holds could be a complicated task. Therefore, to get more intuition, in this subsection we reinstate the AWG model on our single aggregated worker. That is, we assume that for any question $j$, $X_j^A = \mu_j + \epsilon_j$, where the errors $\epsilon_j$ are sampled i.i.d. from $\mathcal{N}(0, \sigma^2)$. For a vector $\vec{Y} \in \mathbb{R}^m$ we denote by $\bar{Y} := \frac{1}{m} \sum_j Y_j$ and $S^2(Y) := \frac{1}{m-1} \sum_j (Y_j - \bar{Y})^2$ its *mean* and its *sample variance*, respectively.

In addition, we assume that the variance estimator $\psi$ is a function of the *aggregated observations* $\vec{X}^A$ (which may or may not be a sufficient statistic for $\sigma^2$), and that all of its directional derivatives exist.

**Theorem 4.3.** *Under the Normal model, for* $m > 3$,

$$\mathcal{R}_{\vec{\mu},\vec{\sigma}}(\text{EB}_A^\psi) = \mathcal{R}_{\vec{\mu},\vec{\sigma}}(A)$$

$$+ \frac{(m-3)^2}{m-1}\Big(E_{\vec{\mu},\vec{\sigma}}\big[\frac{(\psi(\vec{X}^A))^2}{S^2(\vec{X}^A)}\big]$$

$$-2\sigma^2\Big[E_{\vec{\mu},\vec{\sigma}}\big[\frac{\psi(\vec{X}^A)}{S^2(\vec{X}^A)}\big]+E_{\vec{\mu},\vec{\sigma}}\big[\frac{\Sigma_{j=1}^m \frac{d\psi(\vec{X}^A)}{dX_j^A}(X_j^A - \bar{X}^A)}{(m-3)S^2(\vec{X}^A)}\big]\Big]\Big)$$

$$(5)$$

The proof is the full version. Theorem 4.3 derives the explicit risk of the aggregated worker under general dependence structure of $\hat{\sigma}^2$ and $\vec{X}$. The expected reduction in risk when using empirical Bayes i.e., $\mathcal{R}_{\vec{\mu},\vec{\sigma}}(A) - \mathcal{R}_{\vec{\mu},\vec{\sigma}}(\text{EB}_A^\psi)$ can be estimated from the observations, since if $\sigma^2$ is replaced with $\hat{\sigma}^2 = \psi(\vec{X}^A)$ we get an expression which is exclusively dependent on the observations and thus, can be estimated. Corollaries 4.3.1-4.3.2 extend the theorem and demonstrate different conditions which guarantee that EBE will have a lower risk than BLUE.

**Corollary 4.3.1.** *Under the assumptions of Theorem 4.3,*

$$\mathcal{R}_{\vec{\mu},\vec{\sigma}}(\text{EB}_A^\psi) < \mathcal{R}_{\vec{\mu},\vec{\sigma}}(A) \, \forall \vec{\mu} \in \mathbb{R}^m, m > 3$$

*for any* $\hat{\sigma}^2 = \psi(\vec{X}^A)$ *such that:*

$$\frac{E_{\vec{\mu},\vec{\sigma}}\big[\frac{\hat{\sigma}^4}{S^2(\vec{X}^A)}\big]}{E_{\vec{\mu},\vec{\sigma}}\big[\frac{\hat{\sigma}^2}{S^2(\vec{X}^A)}\big] + E_{\vec{\mu},\vec{\sigma}}\big[\frac{\Sigma_{j=1}^m \frac{d\psi(\vec{X}^A)}{dX_j^A}(X_j^A - \bar{X}^A)}{(m-3)S^2(\vec{X}^A)}\big]} < 2\sigma^2$$

$$(6)$$

We promised that under the Normal model we would get more intuition, but Condition (6) is not quite there yet. Note however that if $\psi$ is a constant function (i.e., $\hat{\sigma}^2$ is guessed or estimated not from the data), then a whole chunk of the expression disappears. We next show that this still occurs under a less restrictive assumption.

**Mean-adjusted estimators**  If we use a reasonable variance estimator $\psi$, we would expect a lower estimation as observations are closer to their mean.

**Definition 4.1.** *An estimator $\psi(\vec{X})$ is* mean-adjusted *if for each coordinate $j$ such that $X_j \leq \bar{X}$ (respectively, $X_j > \bar{X}$), we have $\frac{d}{dX_j}\psi(\vec{X}) \leq 0$ (respectively, $\frac{d}{dX_j}\psi(\vec{X}) \geq 0$).*

It is not hard to find estimators that are mean-adjusted, for example $\psi_S(\vec{X}) := S^2(\vec{X})c$ for any constant $c \geq 0$.

**Corollary 4.3.2.** *Under the Normal model, if $\psi$ is mean-adjusted then*

$$\mathcal{R}_{\vec{\mu},\vec{\sigma}}(\text{EB}_{\text{A}}^{\psi}) < \mathcal{R}_{\vec{\mu},\vec{\sigma}}(\text{A})$$
$$+ \frac{(m-3)^2}{m-1}(E_{\vec{\mu},\vec{\sigma}}[\frac{(\psi(\vec{X}^A))^2}{S^2(\vec{X}^A)}] - 2\sigma^2 E_{\vec{\mu},\vec{\sigma}}[\frac{\psi(\vec{X}^A)}{S^2(\vec{X}^A)}])$$

*And hence, $\mathcal{R}_{\vec{\mu},\vec{\sigma}}(\text{EB}_{\text{A}}^{\psi}) < \mathcal{R}_{\vec{\mu},\vec{\sigma}}(\text{A})$ for all $\vec{\mu} \in \mathbb{R}^m$ and $m > 3$ for any $\hat{\sigma}$ which satisfies:*

$$\frac{E_{\vec{\mu},\vec{\sigma}}[\frac{(\psi(\vec{X}^A))^2}{S^2(\vec{X}^A)}]}{E_{\vec{\mu},\vec{\sigma}}[\frac{\psi(\vec{X}^A)}{S^2(\vec{X}^A)}]} < 2\sigma^2 \qquad (7)$$

Now Condition (7) is simple enough to provide some intuition. To see this even better we consider two special cases of estimators:

1. The first special case is when $\hat{\sigma}^2 = \psi(\vec{X})$ is a constant. Then Condition (7) simplifies to $\hat{\sigma}^2 < 2\sigma^2$: for a 'correct guess' $\hat{\sigma}^2 = \sigma^2$ we get the maximal improvement, which becomes weaker as $\hat{\sigma}^2$ drifts towards 0 or $2\sigma^2$.

2. The second special case is $\psi(\vec{X}) = S^2(\vec{X})$. Condition (7) then simplifies to

$$E_{\vec{\mu},\vec{\sigma}}[S^2(\vec{X}^A)] < 2\sigma^2.$$

Since $E[S^2(\vec{X}^A)]$ is itself roughly proportional to the squared error $\sigma^2$ plus $S^2(\vec{\mu})$, we get a valuable indication that Empirical Bayes is expected to perform better on more uniform sets of questions (i.e. whose true answers do not vary substantially).

We provide a proof and show more corollaries in the full version, including the special case of single or multiple workers where $\hat{\sigma}^2$ is a constant.

# 5  EMPIRICAL EVALUATION

In this section, we show experimental results over different datasets and algorithms. We evaluate EBE combined with various truth-discovery algorithms.

That is, we run our $\text{EB}_{\text{A}}^{\psi}$ algorithm, where the baseline truth-discovery algorithm A varies (see below). For the variance

estimator $\psi$ we use the following heuristic, which estimates the variance of each worker using $\vec{X}^A$ as a proxy of the truth, and then takes the average:

$$\psi_H(\boldsymbol{X}) := \frac{1}{n}\sum_{i=1}^{n}\frac{1}{m-1}\sum_{j=1}^{m}(X_{ij} - \bar{X}_j^A)^2.$$

**Algorithms**  We use the following truth-discovery algorithms from the literature: GTM [Zhao and Han, 2012], CATD [Li et al., 2014a], and KDEm [Wan et al., 2016], IPTD [Meir et al., 2021], DTD [Grofman et al., 1983] and, CRH [Li et al., 2014b]. We let each of the aforementioned algorithms up to 14 iterations to converge.

**Datasets**  We used datasets from the following papers: Buildings [Meir et al., 2021] where 208 workers answered 25 questions; Triangles1-Triangles2 [Hart et al., 2018] where 50 workers answered 300 questions; Emotions1-Emotions4 [Snow et al., 2008] where 10 workers answered 200 questions; In addition we have generated synthetic datasets using the AWG model, such that $X_{ij} \sim N(\mu_j, \sigma_i^2)$, where $\sigma_i^2 \sim N(1, 0.5)$. The distribution of the ground truth $\mu_j$ appears on top of each figure.

**Evaluation**  To investigate whether EBE can lower the risk of the above TD algorithms, we sample a subset of workers and questions, run each algorithm and compute the *Improvement Ratio*:

$$IR := \frac{\mathcal{R}(\text{EB}_{\text{A}}^{\psi_H})}{\mathcal{R}(\text{A})},$$

where the (empirical) risk is calculated by taking the average over 1000 samples of $n$ workers and $m$ questions from the dataset.

An *Improvement Ratio* (IR) $< 1$ indicates that Empirical Bayes improves the baseline algorithm A on this dataset.

**Results**  In the synthetic datasets (Figure 2, left) we see that when the ground truth (GT) is constant, EB significantly improves all algorithms, lowering the risk by a factor of 1% - 90%. When the variance of the ground truth is higher (right figures) the IR is closer to 1.

In real world datasets, results are mixed. In the Emotions datasets (Figure 4) there is an improvement, especially when $n$ is low. Figure 6 shows real-world datasets where the ground truth is highly variable, compared to the noise. This high variance causes EB to fail.

However recall that we recommended based on the discussion following Cor. 4.3.2 working with a more 'uniform' sets of questions. To test this point in practice, we partitioned the questions and considered 'uniform' subsets where the variance of the ground truth is low. Indeed, Fig. 3 and Fig. 5 show that on the low-variance datasets, Empirical Bayes improves the outcome and reduces the error.

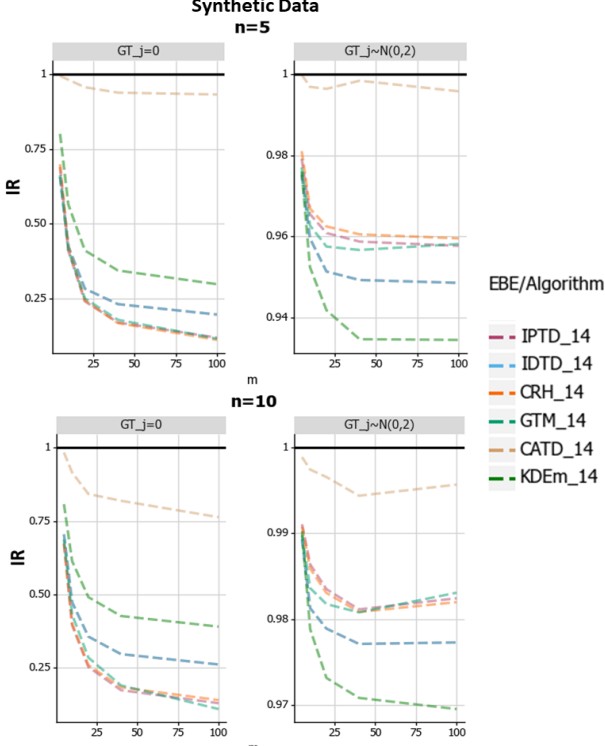

**Figure 2:** IR over synthetic data sets, $n$ is the number of workers and $m$ is the number of questions the left plots are under a constant GT, and the right plots are under a random GT

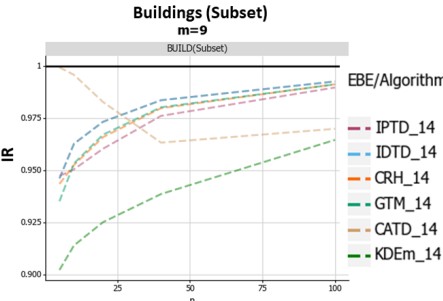

**Figure 3:** IR of a subset of the Buildings dataset reducing the underlying GT variance from 51567 to 633

**To conclude, the results show that Empirical Bayes is particularly effective when there are few workers and low variance of the ground truth, and this applies regardless of the baseline truth discovery algorithm in use.**

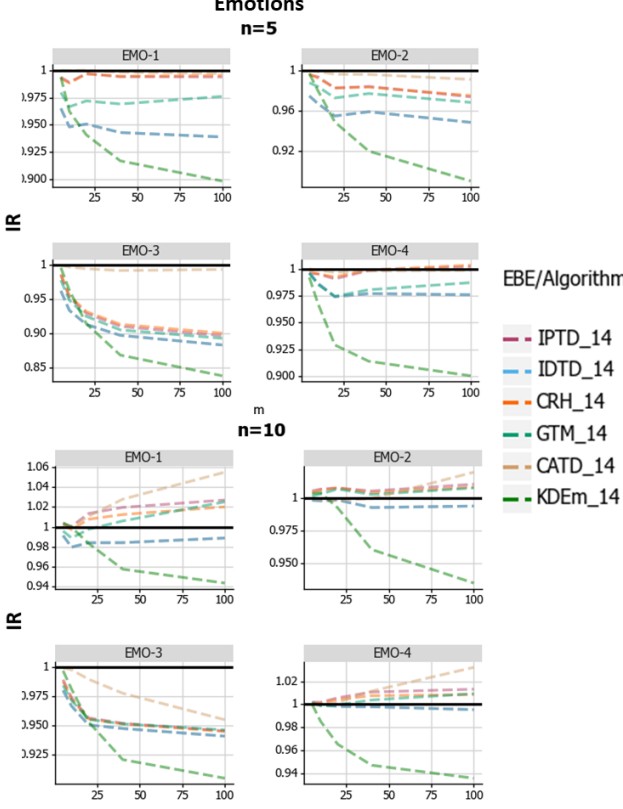

**Figure 4:** $Risk\ Ratio$ for the Emotions datasets

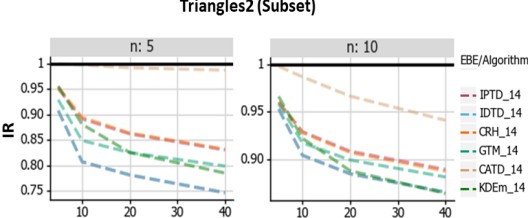

**Figure 5:** IR of a subset of Triangles2 dataset thus reducing the underlying GT variance from 65362 to 195

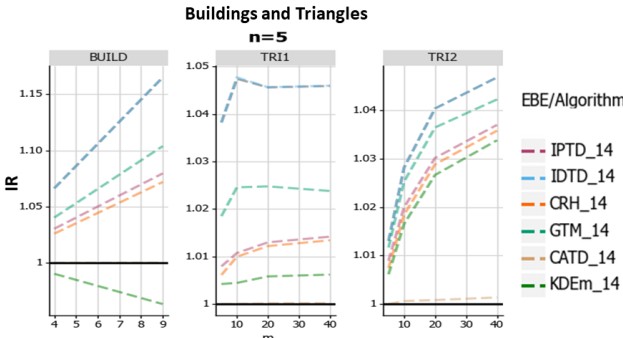

**Figure 6:** $Risk\ Ratio$ for the Buildings and Triangles datasets

# 6 RELATED WORK

As we stated in the introduction, a large part of the truth discovery literature deals with estimating workers' competence. Of those that deal with real-valued data, most make iterative estimations of the ground truth and the competence, and differ in how they implement the steps. For example, Meir et al. [2021] show that workers' average distance between answers from the other workers' answers can estimate their competence; Li et al. [2014a] weigh workers' responses proportionally to the upper confidence interval limit of their estimated variance; Li et al. [2014b] weigh workers' responses by a convex optimization framework which, "minimizes the weighted deviation from the truths to the multi-source input". A different approach taken by Wan et al. [2016] is weighing the responses by the weights which minimizes the kernel density estimation applied to each question separately. Often algorithms use the BLUE estimator with the estimated competences instead of the true workers' variance.

Surveys (such as [Li et al., 2016]) show that there is no single state-of-the-art. Some algorithms work better than others on specific domains and worse on different domains, This highlights the importance of methods that are not algorithm-specific.

A closely related work to ours is of Zhao and Han [2012], where a Bayesian approach is taken. The authors assume prior distributions over the ground truth (Normal distribution), and workers' competence (Beta distribution). Then, an Expectation Maximization (EM) approach is taken for the estimation of workers' competence. The algorithm's output is the posterior mean of the ground truth which incorporates chosen hyper-parameters (prior knowledge) and the estimated competence. We estimate the posterior differently following [Stein, 1956], we do not incorporate any hyper-parameters. Most importantly, our results rely on theoretical foundations.

Other truth-discovery algorithms that deal with binary or categorical labels are outside the scope of this work.

# 7 CONCLUSION

We showed that when workers' competences are known, the Empirical Bayes approach is *always* a better choice (when there are more than 3 questions), and improves *any* TD algorithm that does not have access to workers' competences, for an appropriate variance estimator.

We demonstrated both in theory and in practice that the potential improvement of EBE depends on the uniformity of the set of questions (i.e. it works better when applied to questions whose answers are similar). On the other hand performance also improves when applied to more questions, thus we have an inherent tradeoff between grouping many questions together, or separate them to smaller chunks of 'similar' questions.

Future work might consider how to integrate this into the algorithm, by appropriately partitioning the questions in a way that maximizes the benefit of EBE.

# 8 ACKNOWLEDGEMENTS

This research was supported by THE ISRAEL SCIENCE FOUNDATION (grant No. 2539/20).

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
