# OpenReview forum: "Empirical Bayes approach to Truth Discovery problems"
_auai.org/UAI/2022/Conference — UAI 2022 Poster_

### Official Review · Reviewer_TB69 · 2022-03-24

**Q2(1) Originality/Novelty:** 3
**Q2(2) Significance/Impact:** 3
**Q2(3) Correctness/Technical Quality:** 4
**Q2(6) Clarity Of Writing:** 4
**Q6 Overall Score:** 7
**Q8 Confidence In Your Score:** 4

**Q1 Summary And Contributions:**

This submission deals with the problem of truth discovery through aggregation. It deals with the problem of estimating the true value for different measures provided by agents, where their competence is heterogeneous throughout the measuring. The authors propose a method for answer this when the competence is known, and when it is estimated through the answers provided. Interestingly, the approach is based on reducing the problem to the case with only one agent, who answers with random noise.


**Q2 Assessment Of The Paper:**

More detailed information regarding each of these aspects is given below:

**Q2(4) Quality Of Experiments (Optional):**

3: Good: The experimental evaluation is adequate, and the results convincingly support the main claims.

**Q2(5) Reproducibility:**

3: Good: Key resources (e.g., proofs, code, data) are available and key details (e.g., proofs, experimental setup) are sufficiently well-described for competent researchers to confidently reproduce the main results.

**Q3 Main Strengths:**

The submission is well written and sound. The results advance the area of truth discovery, in particular for multi-agent, multi-question problems.

**Q4 Main Weakness:**

Nothing serious. Perhaps a bit on the comparison of competence between agents and the way these are aggregated.

**Q5 Detailed Comments To The Authors:**

This submission deals with the problem of truth discovery through aggregation. The basic principle
is based on the idea that a set of agents trying to estimate a value are usually *on average*
right. Still, some of the agents may be more competent than others at this task; hence, using a
*weighted average* based on competence may make more sense. The submission deals with the problem
of estimating the true value for different measures provided by agents, where their competence
is heterogeneous throughout the measuring. The authors propose a method for answer this when
the competence is known, and when it is estimated through the answers provided. Interestingly,
the approach is based on reducing the problem to the case with only one agent, who answers with
a random noise.

I found the paper quite interesting. In general, I have always been amazed by the Galton
observations. The issue of handling different questions with agents at different competency levels
is without doubt a hard one, and I was amazed (positively) by the fact that it could be reduced
to the case with only one aggregated agent. The manuscript is clear and well written, and I could
not spot any issues with its technical content. Perhaps my main concern (which, however, does not
affect the main content of the submission) is with the idea that one could measure the relative
competence of different agents; as in the introduction it is mentioned that the opinion of one
agent is worth the opinion of other two agents. I feel that this kind of reading is not meaningful,
but as said, that is outside of the realm of the technical contribution of the submission.

Overall, I feel that this is a nice contribution that could be presented at the conference.

**Q7 Justification For Your Score:**

I believe that, without being an extraordinary paper (hence not with a top score) it is really worth publishing in the conference. The results are nice, and sound.

**Q9 Complying With Reviewing Instructions:**

1: Yes.

---

### Official Review · Reviewer_zf2B · 2022-03-28

**Q2(1) Originality/Novelty:** 2
**Q2(2) Significance/Impact:** 2
**Q2(3) Correctness/Technical Quality:** 2
**Q2(6) Clarity Of Writing:** 1
**Q6 Overall Score:** 4
**Q8 Confidence In Your Score:** 3

**Q1 Summary And Contributions:**

This paper formulates, proves, and empirically tests the conditions for an Empirical Bayes Estimator (EBE) to dominate the weighted mean aggregation. The designed experiments showed that when workers’ competences are known, the Empirical Bayes approach is always a better choice (when there is more than 3 questions), and improves any TD algorithm that does not have access to workers’ competences, for an appropriate variance estimator.

**Q2 Assessment Of The Paper:**

More detailed information regarding each of these aspects is given below:

**Q2(4) Quality Of Experiments (Optional):**

1: Poor: The experimental evaluation is flawed or the results fail to adequately support the main claims.

**Q2(5) Reproducibility:**

2: Fair: Key resources (e.g., proofs, code, data) are unavailable but key details (e.g., proof sketches, experimental setup) are sufficiently well-described for an expert to confidently reproduce the main results.

**Q3 Main Strengths:**

This paper formulates, proves, and empirically tests the conditions for an Empirical Bayes Estimator (EBE) to dominate the weighted mean aggregation. The designed experiments showed that when workers’ competences are known, the Empirical Bayes approach is always a better choice (when there is more than 3 questions), and improves any TD algorithm that does not have access to workers’ competences, for an appropriate variance estimator.

**Q4 Main Weakness:**


--- What’s the meaning of the symbol “_14” in Figures 2-6?

--- The results of UA cannot be seen in Figures 2-6.

--- The authors stated that Triangles1-Triangles12 [Hart et al., 2018] were used in the experiments. However, only the results on the Triangles2 appear in Figure 5 and the results on the Triangles1 (TRI1) Triangles2 (TRI2) appear in Figure 5. Why?

--- In Figures 2-6, the values of n and m are inconsistent and there is no reason for the choice. In addition, since Buildings, Triangles1-Triangles12, Emotions1- Emotions4 are all real-world datasets, the values of n and m in these datasets should be fixed.  How to presents different results with different values of n and m? To my knowledge, only the simulated dataset can be observed the simulation results with different values of n and m.

--- The experimental analysis is somewhat simple. Only the results are presented. More analysis should be analyzed to show the underlying reasons.


**Q5 Detailed Comments To The Authors:**

This paper formulates, proves, and empirically tests the conditions for an Empirical Bayes Estimator (EBE) to dominate the weighted mean aggregation. The designed experiments showed that when workers’ competences are known, the Empirical Bayes approach is always a better choice (when there is more than 3 questions), and improves any TD algorithm that does not have access to workers’ competences, for an appropriate variance estimator.

My main concerns are the experiments and results:

--- What’s the meaning of the symbol “_14” in Figures 2-6?

--- The layout of the current version is very poor. Please present all of the results and figures according to the order of citations in the main body to enhance the readability.

--- The results of UA cannot be seen in Figures 2-6.

--- The authors stated that Triangles1-Triangles12 [Hart et al., 2018] were used in the experiments. However, only the results on the Triangles2 appear in Figure 5 and the results on the Triangles1 (TRI1) Triangles2 (TRI2) appear in Figure 5. Why?

--- In Figures 2-6, the values of n and m are inconsistent and there is no reason for the choice. In addition, since Buildings, Triangles1-Triangles12, Emotions1- Emotions4 are all real-world datasets, the values of n and m in these datasets should be fixed.  How to presents different results with different values of n and m? To my knowledge, only the simulated dataset can be observed the simulation results with different values of n and m.

--- The experimental analysis is somewhat simple. Only the results are presented. More analysis should be analyzed to show the underlying reasons.


**Q7 Justification For Your Score:**

The current experiments and results are not convinced.

**Q9 Complying With Reviewing Instructions:**

1: Yes.

---

### Official Review · Reviewer_kbPk · 2022-04-25

**Q2(1) Originality/Novelty:** 3
**Q2(2) Significance/Impact:** 2
**Q2(3) Correctness/Technical Quality:** 3
**Q2(6) Clarity Of Writing:** 3
**Q6 Overall Score:** 6
**Q8 Confidence In Your Score:** 3

**Q1 Summary And Contributions:**

It presents an approach based on an Empirical Bayes Estimator (EBE) to solve Truth Discovery (TD) problems. The methodology relies on firstly aggregating the workers' answers and then improving this result by applying the EBE. In the case that workers' opinions could be biased or better/worse informed if their competences are input, this will for sure provide better results if the estimator includes this domain-specific info.

The paper provides both theoretical proofs and empirical results.


**Q2 Assessment Of The Paper:**

More detailed information regarding each of these aspects is given below:

**Q2(4) Quality Of Experiments (Optional):**

3: Good: The experimental evaluation is adequate, and the results convincingly support the main claims.

**Q2(5) Reproducibility:**

3: Good: Key resources (e.g., proofs, code, data) are available and key details (e.g., proofs, experimental setup) are sufficiently well-described for competent researchers to confidently reproduce the main results.

**Q3 Main Strengths:**

The paper is well written and the topic is presented in an understandable way, even for non-familiariased readers.

The idea is simple and works fine, it can be integrated with existing algorithms providing better results.

The paper provides theoretical background and empirical support.

**Q4 Main Weakness:**

In the experimentation, the algorithms chosen to use EBE on them are from 2016, the most recent.

The structure is a little bit surprising, offering the related works section almost at the end when the revision of literature generally makes more sense at the beginning to describe the contributions of the presented approach.

Even though the performance is good, I am not totally convinced about the significance of this improvement.

**Q5 Detailed Comments To The Authors:**

The current paper presents an approach based on an Empirical Bayes Estimator (EBE) to solve Truth Discovery (TD) problems. The methodology relies on firstly aggregating the workers' answers and then improving this result by applying the EBE. In the case that workers' opinions could be biased or better/worse informed if their competences are input, this will for sure provide better results if the estimator includes this domain-specific info.

The paper provides both theoretical proofs and empirical results. The practical study is quite interesting to see the performance of the provided approach, using both the perspective of increasing workers (n) and questions (m). In the comparisons given, EBE always improves the baseline algorithm.

- Minor comments:

page 4, a space is missing: " ... risk over4 samples"
the section related work suits generally better in (or after) the introduction to justify the interest of the proposed technique

**Q7 Justification For Your Score:**

I think the idea is interesting enough and it is, in general, well presented and its validity proven both theoretically and with the experiments provided. The code is shared which is always quite positive. The only point to determine is the level of significance of this research result, which is quite subjective and in my opinion, I would say "medium".

**Q9 Complying With Reviewing Instructions:**

1: Yes.

---

### Official Review · Reviewer_6USj · 2022-04-26

**Q2(1) Originality/Novelty:** 2
**Q2(2) Significance/Impact:** 2
**Q2(3) Correctness/Technical Quality:** 2
**Q2(6) Clarity Of Writing:** 2
**Q6 Overall Score:** 3
**Q8 Confidence In Your Score:** 2

**Q1 Summary And Contributions:**

The authors propose an algorithm based on an empirical bayesian estimator in the framework of truth discovery, and compare its performance with that of other existing algorithms.

**Q2 Assessment Of The Paper:**

More detailed information regarding each of these aspects is given below:

**Q2(4) Quality Of Experiments (Optional):**

3: Good: The experimental evaluation is adequate, and the results convincingly support the main claims.

**Q2(5) Reproducibility:**

3: Good: Key resources (e.g., proofs, code, data) are available and key details (e.g., proofs, experimental setup) are sufficiently well-described for competent researchers to confidently reproduce the main results.

**Q3 Main Strengths:**

There is quite some work into the experimental part and the results seem to show the interest and validity of the algorithm.

**Q4 Main Weakness:**

The presentation is very poor, with quite a few typos and unnecessary repetitions. The organisation of the paper is unclear, with a few elements that appear in the wrong order.

With respect to the results, the field of truth discovery is much larger than it is presented here, and there are even Bayesian contributions in somewhat related contexts. The authors should place the paper better within its framework and discuss the connections with the literature.

Some of the technicalities could be made more precise: why is the only main assumption that the estimator is unbiased? Would it be interesting to put some other conditions that guarantee additional results? In spite of the comments by the authors, condition (7) is not too intuitive.

**Q5 Detailed Comments To The Authors:**

The organisation of the paper is quite messy. For some reason, the proofs in the appendix are not given in the order the results appear in the main text. For example, the proof of Theorem 4.3 appears in appendix B.3, and that of earlier Theorem 4.1 appears in appendix B.6. It is as if the authors moved sections around without making a final compilation. The point of the corollaries in Appendix B.5 and C (and why they are placed in that position in the paper) should be discussed in more detail.

In page 4, after Proposition 3.1, it is mentioned 'the following corollaries', but only one is given.

Footnote 3 is a repetition of the earlier footnote 2.

Appendix A.0 is weirdly structured into a number of very short subsections that last only a few phrases each.

Some of the results seem to be given by Casella (as theorem 2.2); in that case you should give a reference to the proof instead of a new (?) one. It is not too clear which are the original contributions in the paper.

Some of the assumptions are too vague not sufficiently justified: why is the only assumption retained in section 4.1 the one of being unbiased? Why is the comparison between the algorithms is made only in terms of the risk defined in page 2?

When presenting the results it is claimed that the performance is good when n is low, but in the figures only the case n=5 is presented. This should be qualified.

Figure 4 is referred to before Figure 3, so it should appear earlier in the text.

Some references are listed but not cited in the text.

Typos:

P1, c1: 'the weight of a the'.
P2, c1: 'vise versa'.
P4, c1: 'We provide the Proof in...'.
P4, c2: Theorem 2.1 should be Theorem 2.2.
P4, c2: 'over4 samples'.
P10: 'In Appendix A We'.
P11: 'is Normally distributed'; 'the Empirical part'.

**Q7 Justification For Your Score:**

The presentation is too deficient and the results seem to be too narrow.

**Q9 Complying With Reviewing Instructions:**

1: Yes.

---

### Decision · Program_Chairs · 2022-05-15

**Decision:**

Accept (Poster)

**Comment:**

Meta Review: This paper is on an interesting/important topic: how to aggregate experts' opinions, on which there's a huge literature. The literature kind of recommends to "average" the opinions in many cases; here the authors show that the average can be improved in some cases with an empirical Bayes approach. The approach is eventually show to be equivalent to the case with only one expert, who answers with some additional random noise.

The reviewers have different opinions on the paper; however there seems to be only one reviewer that is very confident on the paper and who recommends acceptance after an insightful review. The somewhat negative views by other reviewers appear to be mostly based on a too superficial comparison with pre-existing literature/writing to be improved/experiments to be made more convincing.

In my view these negative points, albeit correct, are not sufficient to recommend rejection. The paper makes some nice points on an interesting problem. For this reason I am in favour of acceptance, even though the authors should make an effort to improve the paper in the revised version to take into account the reviewers' comments.